# Effects of multitask training on cognition and motor control in people with schizophrenia spectrum disorders

**Tzu-Yun Chien[1], Jen-Suh Chern [2]\*, San-Ping Wang[1], Yu Yang[3]**

1 Department of Occupational Therapy, Ministry of Health and Welfare Yuli Hospital, Hwa-lien, Taiwan,
2 Graduate Institute of Rehabilitation Counseling, National Taiwan Normal University, Taipei, Taiwan,
3 Department of Occupational Therapy, Tri-Service General Hospital Beitou Branch, Taipei, Taiwan

\* chern8616@gmail.com

**Data Availability Statement:** All relevant data are within the manuscript and its Supporting Information files.

## Abstract

Schizophrenia spectrum disorder (SSD) is a disabling mental illness that causes considerable deficits in motor and cognitive functions. The purpose of this study was to examine the effects of combining traditional multitask training (TMT) and video games--a new form of multitask training (video game multitask training VGMT)--on cognition and motor control performance in people with SSD. This was a quasi-experimental, pretest-posttest design study. A total of 25 patients participated in this study voluntarily (13 males and 12 females, average age = 59.61 years, SD– 11.46 years). All participants underwent two stages of training. The first stage involved TMT, and the second stage involved VGMT. Each training stage was 12 weeks long, with sessions twice a week that lasted for 40 minutes. Cognition, upper extremity motor and postural control performance, and functional mobility and subjective balance confidence were measured at three times: before and after the first-stage training and after the second-stage training. The results revealed that TMT and the combination of TMT and VGMT improved SSD patient's cognition, upper extremity motor control, functional mobility and postural control performance. The subjective confidence of balance during the performance of daily activities was also mildly improved. Training with multitasks in the form of video games tended to further improve the outcome measures. Patients with SSD could benefit from regular participation in various forms of multitasking activities. Whether video games training are better than TMT in improving the functional ability of people with SSD needs further investigation.

**Study protocol registration:** Clinicaltrials.gov, ID: NCT04629898. Registered brief title: Level of Immersion of Virtual Reality and Cognition and Motor Performance in Patients of Schizophrenia Spectrum Disorder.

## Introduction

Multitasking, defined as simultaneously performing more than one task as rapidly and accurately as possible [1, 2] is involved in many real-world activities. It usually occurs in an

**Funding:** This research was funded by Ministry of Science and Technology of Taiwan (https://www.most.gov.tw) grant number MOST-107-2410-H-003-126 and National Taiwan Normal University (https://www.ntnu.edu.tw) grant number106000131.The funders had no role in study design, data collection and analysis, decision to publish, or preparation of the manuscript.

**Competing interests:** The authors have declared that no competing interest exist.

unstructured context. People required nervous system resources, usually sensory-perceptual, cognitive, and motor control resources, to execute adaptive motor control and/or behaviors and to achieve activity goals that are part of their daily routines in the real world [1, 2]. The neuropsychological system receives and processes the inputs that manifest chaotically context, forms a motor behavior plan, and transmits the neurological impulse toward the motor apparatus, which executes motor behaviors [3]. The process is not only critical for multitask performance and maintaining the quality of life in the real world, but also the multitasking in daily life is essential for maintaining and sharpening the functioning of the neuropsychological systems and motor apparatus. Additionally, multitasking requires that the neuropsychological system processes sensory-perceptual and cognitive challenges and makes quick decision based on those inputs to perform adaptive motor and/or behavioral output. During the daily performance of multitasking, unexpected obstacles might arise, exacerbating the challenge faced by the nervous system and changing the efficacy and outcome of task performance [4, 5]. Finally, the implicit order in which the nervous system processes multimodal challenges (including sensory-perceptual and cognitive inputs) and produces motor and behavioral outcomes heavily depends on the characteristics of tasks [6]. Studies have shown that neurocognitive abilities involved in multitasking, including planning, decision making, and execution, which are collectively called multitasking abilities [7, 8]. Multitasking activities are usually goal-oriented. Studies have shown that multitasking as an intervention was effective in improving cognition, motor function, and balance abilities in healthy older adults, patients with balance impairments, older adults with a high risk of falls, and people with stroke, dementia, and Parkinson disease [9–16]. The multitasks used in these studies were usually functional motor tasks (such as walking) with additional cognitive tasks (such as counting down from 100 by 7s). Thus, we proposed that mental effort required for multitasking is higher than that required in single tasking context.

Schizophrenia spectrum disorder (SSD) is a severe mental disease and is prone to relapse, causing disability due to functional limitation [17]. It mainly affects the neuropsychological system. People with SSD initially experience disorganized thought, including hallucination, and delusion, and gradually develop deficits in both basic cognitive abilities (such as attention and memory) [18] and higher-order cognitive functions (such as poor executive function and motor planning) [19], motivation to participate in daily activities, and physical capacity (evident as poor motor efficiency and postural instability) [20]. Executive function is a collective term for cognitive flexibility, inhibitory control, and action planning; the lack of executive function is one of the main causes of poor functional perform in people of neurological conditions. These deficits impair the multitasking abilities of people with SSD, lowering their motivation to participate in the daily activities with multitasking nature and lowering their capability and effectiveness in doing so, leading to functional decline [21]. These deficits might be partially associated with long-term use of antipsychotic medication, which further impedes postural control and increase the incidence of falls [22–24]. The reported incidence of falls in a group of 561 persons with SSD and with a mean age of 53.8 years was 7.1%, which is higher than the fall incidence (5.7%) reported in citizens of the same nation aged between 12 and 64 years old [25]. After a patient experience their first fall, they have a high risk of subsequent falls, hospitalization, declines in physical activity, declined participation in daily multitasking activities such as work activities and poor quality of life, which leads them to withdrawal from independently participating in community activities at a younger age than the healthy aging population, thereby decreases their life expectancy compared with the average population [26]. To halt this downward spiral, effective intervention strategies are needed that target potentially modifiable factors, such as cognitive functions, motor control, and subjective perceptions of falls [27]. Given effects of multitasking training on populations with complex impairment in

motor and cognitive functions, it is reasonable to hypothesize that regular and sufficient training with multitasking activities is beneficial for patients with SSD in terms of mitigating impairment in multiple domains. However, to our knowledge, no study has reported the effects of multitasking training on people with SSD.

Cognitive dysfunction is a core feature of SSD [28]. Impairments in cognitive functions usually occur during the first SSD episode [29, 30]. Moreover, deficits occur across several sub-domains, including attention, working memory, verbal learning, and memory and executive functions. Among these domains, executive functions are the most accurate predictors of functional performance [31]. Several studies have reported that impaired executive functions may also increase fall risks and that fallers performed worse in specific common domains--for example, executive function and attention—than nonfallers [13, 14]. Therefore, multitasking activity interventions that require the individual to devote motor and cognitive effort simultaneously should enable patients to move efficiently and safely [27].

Motor dysfunction has been reported to occur in populations with SSD and to be reported as an early manifestation of this complex disorder [32–36]. Moreover, neuroanatomical findings demonstrated an association between cerebellar dysfunction and structural abnormalities in SSD [37]. Motor abnormalities include decreased upper limb motor proficiency, manual dexterity (measured using the Purdue Pegboard Test), and postural control dysfunction have also been reported [38–41]. In addition, SSD is characterized by an increased and less complex postural sway during challenging sensory conditions, which is likely indicative of cerebellar dysfunction [39–41]. Cerebellar integrity is essential for maintaining proper postural control. Postural control is an intrinsic risk factor for falls and is one of the motor skills for controlling the center of gravity within the base of support (BOS) for maintaining balance. Impaired balance could increase fall incidence [42–45]. Among people with SSD, the magnitude of postural sway increased progressively with an increase in age [44]. Therefore, it is reasonable to argue that the magnitude of postural sway is an appropriated measure of motor control dysfunction in people with SSD. However, to the authors' knowledge, no study has reported the effects of multitasking training on the modulation of postural sway. The present study applied the magnitude of postural sway as an outcome measure of postural control after multitasking training in persons with SSD.

Video games are a new form of multitasking training. Virtual reality technology is used to enable players to devote mental and physical efforts to reach a particular goal in a virtual environment system. Video games are equipped with multiple modes of feedback but not punishment [45], leading to positive user experiences [46]. Therefore, they have been considered to be promising tools for promoting physical fitness, balance, or postural control in several populations [47–50]. However, their potential for improving mental capabilities at fundamental and higher levels was proven only in a population without neuropathological conditions [47]. Moreover, video games were determined to be an acceptable alternative tool for measuring mental state (such as cognition and thought organization) improvement in people with SSD in a study by Campos et al. [51]. Their study, a quasi-experimental trial, examine the effects of video game training on the improvement of cognitive function, lower extremity and trunk motor function, and hand grip strength. The video game used was "Grape Harvest," which requires upper limb, trunk and lower limb stepping movement to be made according to a scenario, and the training protocol consisted of 20-min sessions performed biweekly for 8 weeks. The game was played using Microsoft Kinect equipment. The results showed that the training protocol resulted in no difference in the motor and cognitive function of people with SSD when before and after performance were compared; however, the participants rated the intervention as satisfactory and interactive and reported that regular participation in the intervention led to healthier lifestyles. Moreover, participants considered video games to be as an

acceptable alternative to performing physical activities. The reason for the nonsignificant effects of training on motor and cognitive function might be attributable to the following reasons: (1) inadequate training dosage and (2) insufficient prompting of the exertion of motor and cognitive effort of the participants by the games used. Moreover, whether video game-based intervention with sufficient training time is superior to traditional multitasking activities in terms of multidimensional functions remains inconclusive. Therefore, the purpose of this study was to examine the effects of traditional multitask training (TMT) and combining TMT and video games--a new form of multitask training (video game multitask trainingVGMT)—on cognition and upper extremity motor and postural control in people with SSD. We predicted that the TMT and combining both forms of multitasking training with an increased training dosage would induce significant improvement in motor and cognitive function in people with SSD.

## Materials and methods

All research procedures in this study were reviewed and registered by the Center of Research Ethics of the institute that the corresponding author is affiliated with (Registration number: 2016HM10015).

### Participants

This was a quasi-experimental, pretest-posttest study design. Participants were recruited from a single ward of a 600-bed therapeutic community administrated by a regional psychiatric hospital located in the eastern part of Taiwan. Convenient sampling was employed. Patients participated in this study voluntarily and signed an informed consent form before study commencement. Patients were included if they (1) were diagnosed as schizophrenia or schizoaffective disorder by senior psychiatrists according to DSM-IV criteria [52], with disease onset being >1 year ago; (2) were judged to be symptomatically stable by therapists; (3) were free of interfering or uncontrolled behaviors, (4) could stand independently for at least 3 minutes and walk 10 m without any mobility aids (based on the therapists' observation of participant daily performance); (5) were ≥18 years of age during recruitment for this study. Patients were excluded if they were diagnosed as having musculoskeletal impairment in the past year, had significant visual and auditory impairments, with severe cognitive impairment (based on MMSE score < 18), or were unable to follow verbal instructions or the training and evaluation regimen. The MMSE [53] was administered to all the participants in this study by the same one physician at the facility where the study was conducted one week before the training began.

Details of patient enrollment into the study are presented in Fig 1. In total, 61 patients were assessed for eligibility of whom 21 did not meet the inclusion criteria and 15 patients refused to participate. A total of 25 patients signed the informed consent forms, underwent assessment before intervention, and completed the stage 1 intervention. Three participants dropped out at the end of the stage 1 intervention. Among the patients who had refused to participate in the study initially, three changed their mind and entered the study at the end of stage 1 training. Although they entered this study later, they underwent the same assessment and training procedure. The data of those who dropped out after stag 1 training were excluded from the analysis. Thus, 25 participants (13 male and 12 female) completed two stages of the intervention and three assessment sessions (pretest, posttest 1, and posttest 2), and their data were analyzed. The sample size was calculated before the study was complete. To achieve a statistical power of 0.8, confidence level of 95%, and marginal error of 0.05 in a tow-tailed test with a population size of 61 in the facility where the study was conducted, the required sample size was 27. The

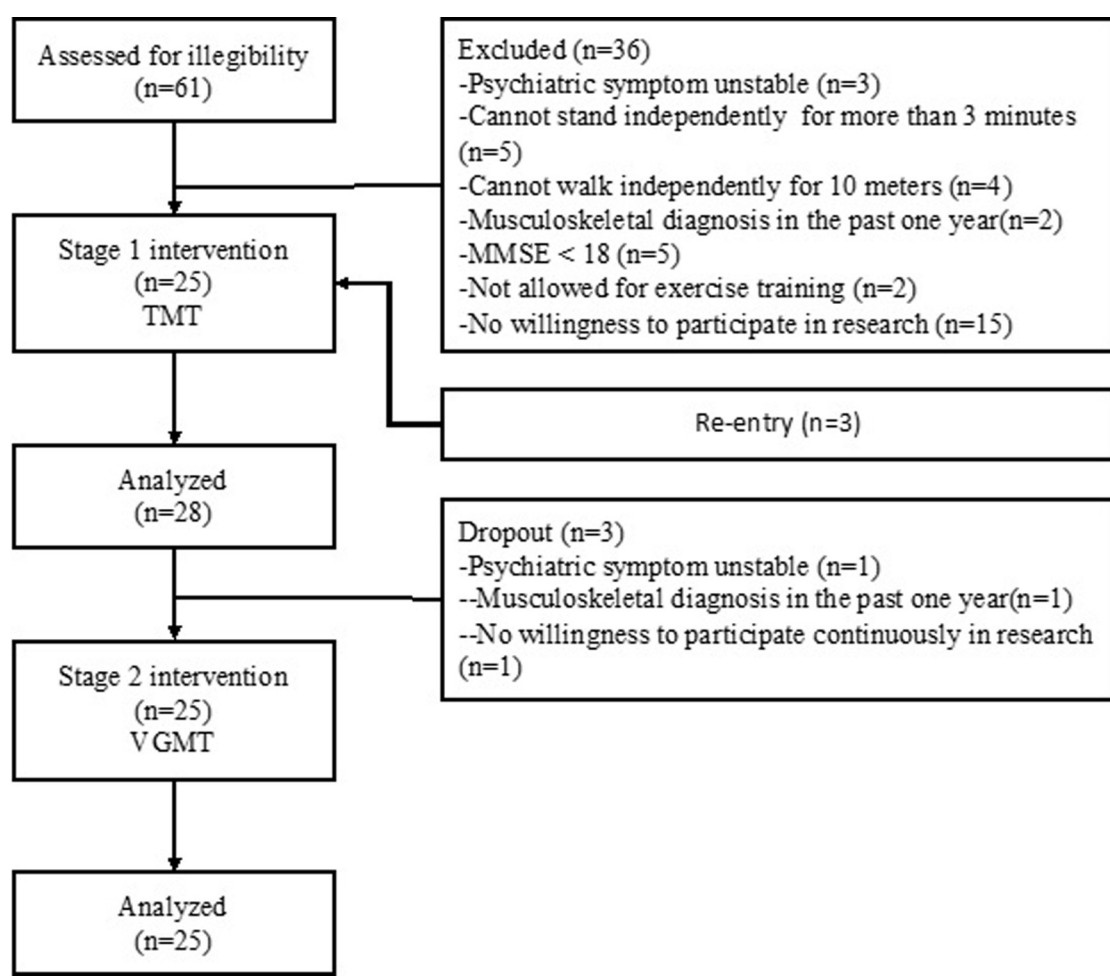

**Fig 1. Participants enrollment and allocation flowchart.**

inclusion of 25 participants in this study might have influenced the data distribution. Therefore, the inferential statistics was adjusted as stated in the "Statistical analysis."

## Procedure

The study procedure is illustrated in Fig 2. First, participants who met the inclusion and exclusion criteria and voluntarily agreed to participate in this study signed the informed consent form. Second, pretraining evaluations were arranged for participants within 1 week before the beginning of the first training stage. Third, a 12-week stage 1 training program was arranged for the participants. Fourth, the first posttraining (Po1) evaluation was conducted within 1 week after the end of the stage 1 training. Fifth, stage 2 training was conducted, which began within 2 weeks after the end of the stage 1 training, lasted for 12 weeks. Finally, the second posttraining (Po2) evaluation was conducted within 1 week of after stage 2 training. Three tests were conducted in total. Each of the training sessions of the two stages was 40 minutes, and each training stage lasted for 12 weeks. Two sessions were conducted in each week. If the participants successfully completed the pretraining evaluation, two-stage training, and two posttraining evaluation as scheduled, the entire study period ranged from 27 to 28 weeks.

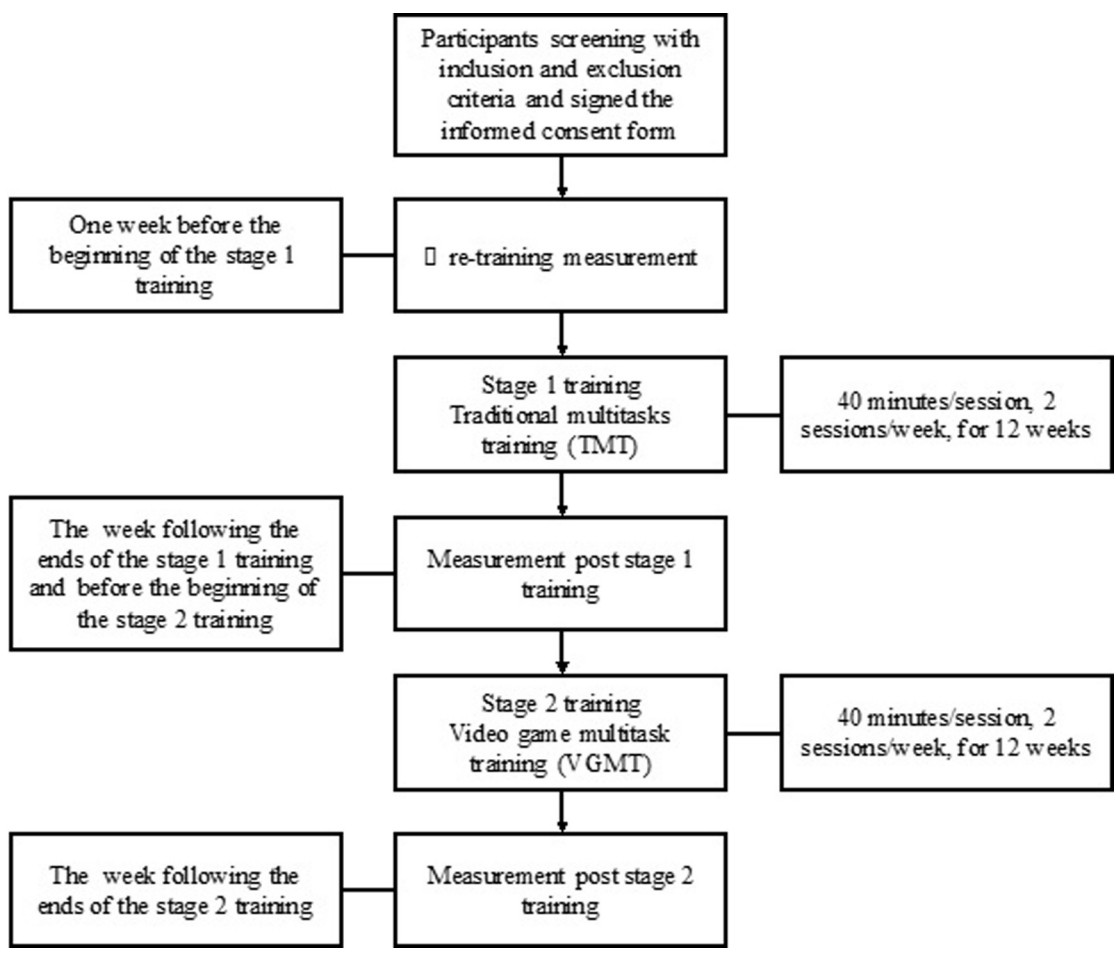

**Fig 2. Study procedure.**

### Training program

As mentioned in the 2.2 Procedure, the training was divided into two stages. Training in both stages comprised different forms of multitasking. The first stage of the training consisted of three real-world and goal-oriented multitasking tasks. **The first task** was the stepping exercise. Participants performed the stepping exercise according to the speed of the metronome. The speed of the metronome was set by the therapist according to the patient's cardiorespiratory endurance to allow the patient to perform the step exercise for 10 consecutive minutes. **The second task** was a beanbag picking-and-throwing activity. A target board for throwing, consisting of five equally spaced holes, was placed 5 m in front of the participants (Fig 3A). The participants were instructed to stand at the initial location with both feet shoulder-width apart, and 10 beanbags were placed on the floor in five locations as shown in Fig 3B. The patient was allowed to choose which beanbag to pick and throw first. The patient picked up one beanbag at a time by bending the trunk and with both feet firmly on the ground. After returning to the full upright standing posture, the participant threw the beanbag toward the hole after a lighting prompt. Each patient was asked to complete three runs of beanbag picking-and-throwing in 15 minutes. Each run involved picking-and-throwing 10 beanbags. **The third task** was ball kicking. The participants stood at the designated initial location. Five volleyballs were placed at an equal distance at 5 m in front of them, and five wooden cases with the openings toward

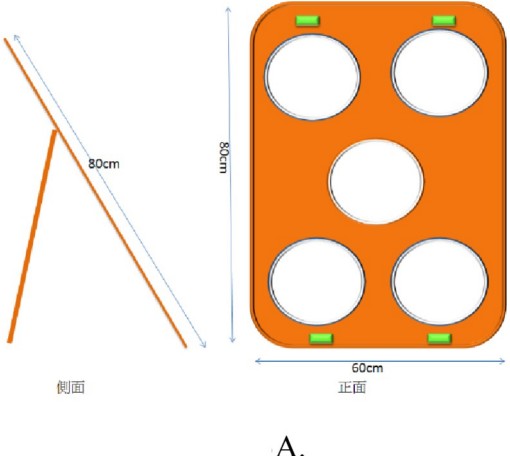

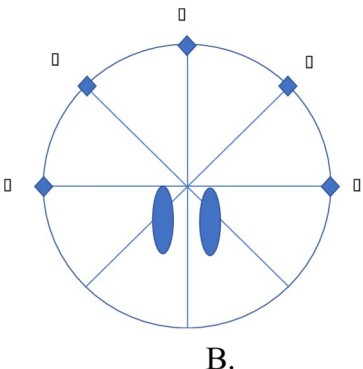

**Fig 3.** A. The design and outlay of the beanbag toss board. B. A-E indicates the location of the beanbags. There are two beanbags at each location. The two ovels are the location where the feet were placed when the participants performed the task.

the participants were placed in front of the participants 10 m away. To begin the ball kicking activity, the participants walked at their own speed toward any one ball and kicked the ball into the corresponding wooden box. The participant then walked to the position of the next ball and kicked it. The participants had to return to the initial position after all five balls were kicked. The balls were returned to their original positions for the next run of kicking. Each participant had to complete six runs; thus, each participant kicked a total of 30 balls in 15 minutes. A short period of resting (approximately 1–2 minutes), during which the participant could sit in a chair, was granted whenever a run was completed during task performance at the participant's request. Therefore, the training time in each session was the same across participants: 40 minutes. Due to limited space and equipment, all the interventions in stage 1 were performed on an individual basis.

The three tasks selected in the first stage were goal-directed tasks that occupational therapist often use as intervention modalities, to improve patients' functional performance. Performing such tasks requires the participant to demonstrate motor responses according to the cognitive and motor demands of the tasks. Therefore, the authors considered that tasks to be real world multitasking activities, However, no formal research has reported their effects on people with SSD.

Stage 2 training consisted of two sports video games, also known as exergames. The video game console used is Xbox-Kinect, and the virtual environment of the game was displayed on a 50-inch LED monitor. The console was placed on a table at a height of approximately 50 cm and at a distance of 5-m away in front of the player, and the players' viewing angle was between ±10 degrees. The games selected were tennis hit-back games and golf targeting games. In the tennis hit-back game, the player was required to stroke back the ball from the server in the game with a forehand or a backhand shot. If the stroked ball hits the doll in the virtual environment, cheering sounds were emitted from the gaming machine, and the time for hitting back was extended, which was displayed on the monitor as visual feedback. For a real-world effect, each participant was provided a real tennis racket to be held with their dominant hand. Before the games began, the trainer instructed the participants in the correct method of holding the racket and performing the hitting-back motion to avoid sports injury. During the game, the trainer instructed the players to move their feet or shift their body's center of gravity in correspondence with the direction of the incoming balls to increase the chance of successfully hitting the doll in the virtual environment. This was to motivate the participants and to provide them with a sense of accomplishment for continuous participation. In the golf shot game, the participant was required to strike the ball toward the target in the virtual environment in a manner similar to how it is done in a real golf game. Each round involved striking 10 balls; when the ball hits on the dart target, the display score designated for the hit position appeared as visual feedback to the player. This game has no other forms of feedback except visual feedback. To increase the realism, the trainer prepared a straight-grip trekking pole for the participants to simulate the manner of holding a golf club (non-dominant hand on the top of the dominant) for hitting the ball. In addition, before the game started, the trainer taught the participant how to hold the club and instructed the participant to stand at an appropriate position relative to the display screen of the game; once the game started, the trainer guided the participants verbally to adjust their standing position and body posture to allow the ball to hit the location designated with a high score to increase the motivation and sense of accomplishment of the player. In the stage 2 training, each video game lasts for 20 minutes; each run of the tennis game lasted 4 minutes, and each participant was required to complete five rounds of the game; the golf game required 5 minutes per round, and each participant completed four rounds. The stage 2 intervention was also conducted on an individual basis.

The reason for using the two video games was that both have goal-oriented multitasking characteristics, requiring not only the participants' exertion of cognitive resources but also appropriate movement responses of the upper and lower limb and the trunk based on the cognitive information inputs. We, therefore, believed that regular and frequent participation in these tow video games could induce learning effects of cognitive and motor function in the participants.

## Measures

The first author was responsible collecting demographic data and outcome measurement. The demographic (including age, sex, body height, and body weight) and background data (MMSE, activities of daily living, instrumental activities of daily living, and participation in vocational training) were collected. The MMSE was used as a screening tool [53] to ensure that the participants had adequate cognitive ability to participate in this study, and activities of daily living and instrumental activities of daily living measurements using the Barthel Index was used as a background data. The training outcome was measured using the following outcome measures:

**Activities-specific balance confidence scale.** The Activities-Specific Balance Confidence (ABC) Scale is an 11-point self-reported scale comprising 16 items. The participants were

required to rate their level of self-confidence in maintaining their postural balance while performing various activities, such as walking around a living space, walking up or down the stairs, reaching for a small can off a shelf at eye level, walking up or down a ramp, and being bumped into by people as they walk around a shopping mall. Each item was rated as a whole number (0–100). The participants' ABC Scale scores were derived by dividing the total score (0–1600) by 16.

The ABC Scale is a measure of objective balance and functional mobility. Its internal consistency in community-dwelling older adults was excellent (Cronbach's alpha = 0.96) [54], and the test–retest reliability for an elderly population was also excellent ($r = 0.92$, $p < 0.0001$) [55]. Furthermore, the ABC Scale was highly correlated with the Fear of Falling Avoidance Behavior Questionnaire ($r = 0.678$, $p < 0.001$), indicating excellent construct validity [56].

**Box and block test.** The Box and Block Test (BBT) measures unilateral gross manual dexterity of the upper limb and requires the participant to perform simple grasp-and-release movements. A test box with 150 blocks and a partition in the middle is placed lengthwise along the edge of a standard-height table, and the participant sits on a standard-height chair facing the box. The participant is instructed to pick up one block at a time with the right (or left) hand, carry it to the other side of the box, and drop it into the compartment at the other side as soon as the fingertips cross the partition. The size of the block is 2.5 cm × 2.5 cm × 2.5 cm, and the height of the partition is 30 cm. The score is the number of blocks carried from one compartment to the other in 1 minute. Each hand was scored separately. The norm and psychometric characteristics have been established for the elderly people and people with stroke [57, 58]. As well as we know, this is the only study to measure the gross manual dexterity of the upper limb by using the BBT in people with SSD as an outcome measure after a formal intervention program.

**Color trails Tests 1 and 2.** The Color Trails Test (CTT) is a language-free and culture-fair instrument version of the Trail Making Test [59] and was developed to allow for the assessment of sustained and divided attention and executive function in adults. It comprises two tests, CTT1 and CTT2. CTT1 is administered first and requires the respondent to connect circles in an ascending number sequence between 1 and 25; CTT1 is followed by CTT2, which requires the participant to connect numbers in an ascending sequence while alternating between pink and yellow colors. The same numbers are presented twice, once in pink and once in yellow. Therefore, the participant must ignore the distracter items (for example, number 3 in a yellow circle) and connect the correct number in a pink circle. The time taken to complete each part of the CTT1 is recorded in seconds. Qualitative aspects of the performance, including near misses, prompts required, and sequencing error in color and number, which were indicative of brain dysfunction were recorded for each task. Neuropsychologists believe that the CTT requires the operation of higher-level cognitive ability to form an efficient upper extremity motor plan [59]. The test results indicate execution efficiency of daily multitasking activities, such as driving [60].

**Timed Up and Go Test.** The Timed Up and Go Test (TUG) is an assessment of functional mobility, including balance, sit-to-stance transfer, and walking. It was initially designed for detecting fall risks in elderly persons. However, it has been used in people with central nervous system pathologies, such as Alzheimer disease and stroke, and in those with lower limb orthopedic conditions, such as hip fractures [61, 62]. Patients were instructed to start the test in a seated position. On the verbal command"begin," the participants had to stand up from the chair, walk ahead for 3 m, turn around, walk back to the chair, and sit down. The time required by the participants to finish the task was the performance score and was measured in seconds. Both intratester and intertester reliability are high in populations with cognitive dysfunction [63].

**Center of pressure-related measures.** Center of pressure (COP)-related measures are widely accepted measures of control of COP movement, which reflects the efficiency of the central mechanism to control the amount of COP shifting during either static standing or dynamic task performance [64]. During static standing, bilateral feet placement constructs the configuration of base of support and changes the measured COP as a whole (Total Path Length, TPL) in the mediolateral direction (COPX) or in the anteroposterior direction (COPY) due to the adaptive neuromechanism in correspondence with the perceived base of support. The larger the COP-related measures are, the less steady the performers is, indicating the adaptive neuromechanism has deceased efficiency and the individual has poor standing balance [65]. COP-related measures in this study were obtained using the Footscan Pressure Measurement System equipped with a three-dimensional interface box (with a data acquisition rate of 50 Hz) and a 0.5 m pressure measurement mat (with 4096 active sensors).

First, the participants were instructed to stand as steady as possible on the pressure measurement mat with 2 (eyes open vs. eyes closed) × 3 (bilateral feet shoulder width apart stance, bilateral feet side-by-side stance, and tandem stance) = 6 stance conditions. The two-dimensional COP coordinates were recorded for three 10-second trials, and COPTPL, COPX, and COPY were calculated using Balance 7.4. The average of the three trials in each stance was divided by the leg length in mm to represent the standing balance and was used for statistical analysis.

Second, the COP measurements performed while standing were also performed while the participant performed the Functional Reach Test (FRT). The FRT is a single-item test developed as a quick screen for dynamic balance problems. The original testing procedure is as follows: (1) a yard stick is attached to the wall at shoulder height; (2) the participant is instructed to stand with feet shoulder-width apart, make a fist, raise the arm up to a shoulder flexion of 90˚, and reach forward with the trunk bending forward as much as possible against the yard stick on the wall with both feet firmly placed on the ground and both knees straight. The initial reading is subtracted from the final to obtain the functional reach score.

This study did not use the functional reach score because it might not be as sensitive as COP-related measures in detecting changes in the neuromechanism for postural control [66]. Therefore, COP-related measures were obtained while the subject stood on the pressure measurement mat and performed the FRT. COPTPL-FR, COPXFR, COPYFR during the FRT were hypothesized to be a precise measure of the control of dynamic balance. Furthermore, the measures were divided by the body height to normalize the impact of the pendulum radius on the COP swing amplitude. In dynamic conditions, the larger the measures are, the better the dynamic balance is.

## Statistical analysis

SPSS 23.0 statistical analysis software package (IBM, Chicago, USA) was used for all inferential statistics. The background data of the participants were analyzed descriptively. The differences in the outcome measures at three testing time points, namely prettraining test (Pre), posttraining test after the stage 1 training (Po1), and posttraining test after stage 2 training (Po2), were examined with the repeated measures analysis of variance (ANOVA). Before determining significant through repeated-measures ANOVA, the spherical hypothesis, which is the normal distribution of the data, was examined using Mauchly's test, When the spherical hypothesis was violated, the Greenhouse—Gaiser test, in which the degree of freedom is adjusted for the calculation of $p$ values, was used. The default pairwise comparison with Bonferroni adjustment was used for post hoc comparison when significant differences were noted between three testing time points. Furthermore, ηp2 was calculated as the effect size. The significance level was

set at $p < 0.05$; $\eta_p^2 < 0.058$ indicated a small training effect, $0.058 \leq \eta_p^2 < 0.138$ indicated a moderate training effect, and $\eta_p^2 \geq 0.138$ indicated a large training effect.

## Results

Table 1 presents the participants' demographic and clinical characteristics. The mean age of the participants was 59.61 years, indicating aging of the participants. MMSE scores ranged from 20 to 33, with average scores of 26.84 indicating moderate cognitive ability. Over half of the participants joined the vocational training program. MMSE scores showed that some participants had prominent cognitive deficits, whereas others did not. ADL and IADL scores indicated that participants could independently perform daily activities.

Table 2 presents a summary of the results of repeated-measures ANOVA for motor and cognitive functions and functional mobility. Results revealed that CTT1 T, CTT2 T, and CTT2 Cue were significantly different among three testing time points; the effect sizes for all measures of CTT1 and CTT2, except CTT2 Cue, were moderate to large ($\eta_p^2 = 0.058$–$0.198$). Post-hoc pairwise comparison revealed that both traditional multitask training (TMT) and video game training improved the scores of CTT1 and CTT2; multitask training in the form of sports video games could further extend or improve the training effects on cognition ($p < 0.05$–$0.01$).

The results further showed that traditional multitask training (TMT) followed by sports video game training (VGMT) nonsignificantly affected ABS score ($p = 0.180$) with a moderate effect size ($\eta_p^2 = 0.074$). Conversely, the TUG scores at three measured time points were significantly different ($p < 0.01$), with a large effect size ($\eta_p^2 \geq 0.138$), and the pairwise comparison indicated that VGMT following TMT further improved TUG scores significantly ($p < 0.01$).

Finally, Table 2 shows that the BBT of the dominant (or right) hand was affected by two forms of multitask training ($p = 0.022$), especially the form of video games following traditional multitasks ($p < 0.05$). Generally speaking, the continuous training of two types of multitasks imposed a large effect ($\eta_p^2 \geq 0.138$) on the gross motor function improvement of the dominant hand.

Table 3 shows the training effects on COP-related measures. The results revealed that the majority of all COP-related measures during static standing were significantly different at the three measured time points except the TPL during standing with bilateral feet shoulder width apart (BSOTPL) ($p < 0.01$) with a large effect size ($\eta_p^2 = 0.209$–$0.913$). The pairwise comparison showed that multitasking in the form of video game following traditional multitasks could further improve COP measures compared with the TMT.

**Table 1. Demographics of the participants.**

| | Average ± SD | Ranges | Ratio |
|---|---|---|---|
| sex(M:F) | | | 12:13 |
| Age(yrs) | 59.61 ± 11.46 | 35.03–83.42 | |
| BH(cm) | 161.51 ± 7.40 | 146–176 | |
| BW(Kg) | 64.29 ± 14.76 | 39–98 | |
| MMSE | 27.40 ± 3.79 | 20–33 | |
| ADL | 99.57 ± 1.3 | 895–100 | |
| IADL | 14.91 ± 5.2 | 25–22 | |
| Participated in Vocational Training (No:Yes) | | | 9:16 |

BH: Body Height, BW: Body Weight, MMSE: Minimal Mental Status Examination, ADL: Activity of Daily Living, IADL: Instrumental Activity of Daily Living, Ranges: minimum to maximum value

**Table 2. Summary of repeated measure analysis of variance in cognition, upper extremity gross motor function, functional mobility and confidence of falls.**

| Task | | Pre | Po1 | Po2 | $F$ | $p$ | $\eta_p^2$ |
|---|---|---|---|---|---|---|---|
| | | n = 25 | n = 25 | n = 25 | | | |
| CTT1 | T | 124.42 ± 91.85 [a] | 110.04 ± 70.15 [a] | 91.75 ± 41.24 [b] | 4.008 | .025* | .143** |
| | CE | .65 ± 1.25 [a] | .69 ± 2.03 [a] | .14 ± .27 [b] | 2.224[#] | .132 | .085** |
| | NE | .17 ± 0.47 | .09 ± .28 | .05 ± .07 | .851[#] | .234 | .058* |
| | Cue | 1.53 ± 3.16 | .96 ± 2.01 | .52 ± .73 | 2.395[#] | .121 | .091* |
| CTT2 | T | 237.61 ± 139.34 [a] | 212.90 ± 106.24 [a] | 174.85 ± 55.20 [b] | 5.970 | .005** | .198** |
| | CE | 2.42 ± 4.36 [a] | 1.38 ± 2.86 [a] | .43 ± .58 [b] | 3.884[#] | .038* | .139* |
| | NE | .34 ± .55 | .24 ± .571 | .48 ± .77 | .978 | .383 | .039 |
| | Cue | 4.72 ± 6.41 | 3.51 ± 3.73 | 2.15 ± 1.76 | 3.415[#] | .041* | .125* |
| ABC | | 76.54 ± 20.18 | 82.45 ± 15.61 | 82.02 ± 15.70 | 1.910[#] | .180 | .074* |
| BBT | R | 49.36 ± 12.70 [a] | 53.41 ± 11.06 [a] | 56.12 ± 12.81 [b] | 5.466[#] | .022* | .185** |
| | L | 50.64 ± 11.60 | 52.21 ± 10.88 | 52.10 ± 11.49 | .832[#] | .394 | .034 |
| TUG (sec) | | 10.94 ± 2.16 [a] | 10.59 ± 1.95 [b] | 9.52 ± 1.92 [c] | 33.478[#] | < .001*** | .593* |

*$p < .05$ or $.058 \leq \eta_p^2 < .138$;

**$p < .01$ or $\eta_p^2 \geq .138$; CTT: Color Trail Test, T: Time in second, CE: Color Error, NE: Number Error, ABC: Activities-specific Balance Confidence scale, BBT: Box and Block Test, R: Right arm, L: Left arm, TUG: Timed Up-and-Go Test. The same superscript [a], [b], [c]: The same superscript: nonsignificant difference after post hoc comparison; the different superscript [a], [b], [c]: significant difference after post hoc comparison. The superscript

[#]: F calculated by adjusted degree of freedom.

Conversely, only COPTPL in the dynamic context (FRT) among the three testing time points was significantly different ($p < 0.01$) with a large effect size ($\eta_p^2 \geq 0.138$). Pairwise comparison revealed that COPTPL during the FRT at Po2 was not significantly different from that at Po1. COPX during the FRT was not significantly different among the three testing time points, but a moderate effect size was found. COPX during the FRT at Po1 increased as compared with Pre but decreased in Po2 as compared with Po1 and Pre.

## Discussion

This is the first study to explore the effects of two types of multitask training—TMT and VGMT—on cognitive function, upper limb motor ability, functional mobility, and dynamic and static postural control as well as subjective perception of balance in people with SSD. All training outcomes were measured using reliable and valid instrument. Functional mobility was measured objectively by using the TUG and subjectively by using the ABC. The results showed that two consecutive training stages, with each stage comprising 12 weeks of training, two sessions per week, and, each session lasting 40 minutes, had significant effects on the majority of the outcome measures. In addition, multitasking in the form of video games following traditional multitask training in stage 1, which was conducted at the stage 2 training, tended to extend or further improve the effects of TMT on the majority of the outcome measures, especially on those representing the postural control as determined by COP-related measures.

First, training people with SSD with both forms of multitasking (traditional multitasking followed by video games) significantly improved their performance on the CTT with a large effect size. The training effects might result from the multitasks nature of the CTT. The CTT requires a participant to use cognitive ability to judge the perceptual information (including color and number sequence) input while performing the motor skill (that is, holding a pen and tracing the trail). Therefore, the authors believe that the tasks in CCT are multitasks in nature.

**Table 3. Summary of repeated measure analysis of variance in center of pressure control during static stance and dynamic weight shifting.**

| Task | COP | Pre | Po1 | Po2 | $F$ | $p$ | $\eta_p^2$ |
|---|---|---|---|---|---|---|---|
| | | n = 25 | n = 25 | n = 25 | | | |
| BSO | TPL | .039 ± .050 | .029 ± .012 | .037 ± .033 | .488 | .565 | .020 |
| | X | .002 ± .001 [a] | .002 ± .001 [a] | .001 ± .000 [b] | 9.093 | < .01** | .275** |
| | Y | .002 ± .001 [a] | .002 ± .001 [b] | .001 ± .000 [c] | 22.981 | < .01** | .489** |
| BBO | TPL | .035 ± .012 [a] | .031 ± .010 [a] | .022 ± .008 [b] | 15.816 | < .01** | .397** |
| | X | .004 ± .002 [a] | .003 ± .001 [a] | .002 ± .000 [b] | 16.264 | < .01** | .404** |
| | Y | .003 ± .001 [a] | .003 ± .001 [a] | .002 ± .001 [b] | 18.188 | < .01** | .431** |
| TDO | TPL | .066 ± .020 [a] | .004 ± .002 [b] | .003 ± .001 [c] | 250.392 | < .01** | .913** |
| | X | .484 ± .355 [a] | .004 ± .002 [b] | .003 ± .001 [c] | 45.798 | < .01** | .656** |
| | Y | .005 ± .002 [a] | .003 ± .001 [b] | .002 ± .001 [c] | 21.816 | < .01** | .476** |
| BSC | TPL | .029 ± .007 [a] | .031 ± .013 [a] | .020 ± .007 [b] | 8.372 | < .01** | .259** |
| | X | .001 ± .001 | .002 ± .001 [a] | .001 ± .001 [b] | 6.343 | < .01** | .209** |
| | Y | .002 ± .001 [a] | .002 ± 0.001 [a] | .001 ± .000 [b] | 12.417 | < .01** | .341** |
| BBC | TPL | .041 ± .018 [a] | .034 ± .010 [a] | .025 ± .012 [b] | 10.829 | < .01** | .311** |
| | X | .004 ± .002 [a] | .002 ± .001 [b] | .002 ± .001 [b] | 24.804 | < .01** | .508** |
| | Y | .003 ± .001 [a] | .003 ± .001 [b] | .002 ± .001 [c] | 21.069 | < .01** | .467** |
| TDC | TPL | .081 ± .031 [a] | .099 ± .063 [a] | .043 ± .019 [b] | 17.228 | < .01** | .418** |
| | X | .005 ± .004 [a] | .006 ± .004 [a] | .002 ± .001 [b] | 8.521 | < .01** | .262** |
| | Y | .005 ± .002 [a] | .004 ± .003 [a] | .002 ± .001 [b] | 14.001 | < .01** | .368** |
| FRT | TPL | .052 ± .014 [a] | .071 ± .025 [b] | .073 ± .025 [b] | 14.724 | < .01** | .380** |
| | X | .004 ± .001 | .005 ± .002 | .005 ± .002 | 2.315 | .110 | .088* |
| | Y | .010 ± .003 | .010 ± .002 | .011 ± .003 | .853 | .432 | .034 |

*$p$ < .05 or .058≤$\eta_p^2$ < .138;

**$p$ < .01 or $\eta_p^2$≥138; COP: Center Of Pressure, BS: Bilateral feet shoulder width stance, BB: Bilateral feet side-by-side stance, TD: Tandem stance, O: Eyes Open, C: Eyes Close, TPL: Total Path Length, X: COP coordinate in medial-lateral direction, Y: COP coordinate in anterior-posterior direction, FRT: Functional Reach Test. The same superscript [a], [b], [c]: nonsignificant difference after post hoc comparison; the different superscript [a], [b], [c]: significant difference after post hoc comparison.

The aforementioned result indicated that both forms of training tasks in this study might be effective in resolving the reported difficulties in functional performance when people with SSD face an unfamiliar and complex multitasking situation [2]. Multitasking difficulties in these population were related to complex cognitive functions across multiple domains, including cognitive flexibility, inhibitory function, planning ability, prospective memory and attention shifting as measured by CTT. These multifaceted cognitive functions are collectively called executive function [67]. In this study, the CCT was used as a measure of executive function, a fundamental measure of multitasking abilities. The results of this study support those of previous studies that determined that tests with multitasking characteristics should be used to detect the heterogeneity of cognition in people with SSD and as a crucial outcome measure of multitasking abilities [67]. In addition, the study results indicated that a 12-week multitasking training program, comprising 40-minute VGMT sessions conducted twice weekly, might further enhance the multitasking abilities of people with SSD after the same dosage of TMT. VGMT provides a virtual multitask scenario in which participants need to integrate different domains of cognitive abilities as an important basis for planning and encouraging the motor apparatus to execute physical and motor responses. Although our results showed significant improvement in patients' performance in the CTT, previous findings on the dissociation between the multitasking test performance and real-world multitasking performance [68, 69] indicates that people with SSD are likely to have difficulty applying their training-acquired multitasking

ability in real-world multitasking activities. Follow-up studies measuring functional performance before and long after training are necessary.

Another notable finding is that the gross motor function of the dominant arm measured using the BBT in people with SSD significantly improved, especially after the stage 2 VGMT. Furthermore, studies have reported an association between upper limb motor deficiency and the performance of daily multitasking activities in people with SSD. However, to our knowledge, no study has reported on the clinical effort devoted to upper limb motor function training in SSD. The results may imply that the benefits of the training program in this study on cognition may produce effects that could spill over to other abilities that are closely related to cognitive abilities [70]. The tasks in the stage 1 training mostly required automatic associated lower limb and trunk movements to maintain the balance and safety in dynamic situations, and the mild amount of upper limb involvement in the beanbag toss activity might have improved upper limb motor function at Po1. This trend may have been further strengthened during the stage 2 VGMT, which required the involvement of the dominant upper extremity. The study results revealed difference in the cognitive and upper limb motor function between pretraining and posttraining in the studied population.

Another key finding of this study is that two consecutive stages of different types of full-body physical activity training with multitasking traits yielded prominent benefits for people with SSD in terms of dynamic and static postural control and functional mobility. This finding has important implications for preventing the fall risk in people with SSD who have lived in therapeutic communities for a long time. Long-term clinical observation and investigation have revealed that people with chronic SSD living in an institution have a 0.2% chance of falling, and falls cause further disability and may result in fatality in patients with poor functioning [71]. Clinical staff usually find it difficult to involve people with SSD in specially designed training activities for improving posture control because of the patients' low motivation to participate and level of participation, possible due to the illness itself or the side effects of medications used in treating the illness. Thus, the effects of traditional exercise (such as jogging and walking) on decrease in fall incidence and improvement of functional performance tended to be limited. The study findings may be influenced by the motivation and participation level induced through interesting and appropriate cognitive and whole-body physical activity challenges and the multitasking nature of training tasks. The improvement of functional mobility, which required the involvement of dynamic posture control, and standing stability, which required the involvement of static postural control, represented the improvement in the body's balance performance. Thus, VGMT following TMT might be able to reduces long-term fall incidence and improves the functional performance in productive activities [72] in people with SSD. Future research is warranted to evaluate the effects of this training type on the decrement of fall incidence in people with SSD.

Interventions to improve the posture control and cognitive functions in people with SSD have been studied separately. The present study found that training with either TMT or VGMT could simultaneously improve cognition and posture control in people with SSD, indicating that postural control is associated with cognition. The devotion of cognitive resources for controlling postural stability (measured by the amount of COP shift) makes it a voluntary control task instead of a simple automatic response. During nonroutine daily activities, the cerebral cortex participates at all levels. The result supported the hypothesis of Mullick et al. [70] that cognition and motor performance in subjects were associated with central nervous system injury or illness in people with SSD. Future research is necessary to determine the effect of the multitasking full-body training model used in this study on the structure of the prefrontal area and motor cortex.

The ABC is a subjective measure of confidence in maintaining balance when performing daily activity. This study failed to detect the training effects on the ABC measures, indicating

that people with SSD may not experience changes in their balancing capacity corresponding to the microscopic improvement in postural control. The participants were those with chronic illness who have lived at the therapeutic community for a long time and hence are familiar with the environment, which might have played a role in increasing their confidence in managing daily environmental challenges. The patients have the opportunity to go out shopping once a week, it happened on a weekday when there are few people present; consequently, the environmental challenges they experience are limited. The subjective measurement of confidence in maintaining balance should be conducted in an unfamiliar situation to determine the improvement.

Finally, the study results suggested that both multitasking in the real world and in video games are effective training modalities for improving cognition, motor function, and posture control in people with SSD. In this study, participants received TMT first for 12 weeks followed by VGMT for another 12 weeks, and the time interval between the two training programs was only two weeks. All the outcome measured at Po2 tended to be better than those measured after the stage 1 training (Po1) and before training (Pre). Although not all the different outcome measures did not significantly differ at the three measured time points, the training effect size exceeded the moderate level. Besides, video games might be favored by the participants for the following reasons [51]: (1) Video games have three-dimensional virtual scenarios, allowing players to experience rich situations in a safe context and to satisfy their adventure instincts. (2) The program itself is designed with multisensory feedback, which provides users a sense of authenticity and accomplishment, thus encouraging users to continue use, and leading to a high degree of physical and mental investment. (3) The no-penalty system design allows users to practice various cognitive and motor integration strategies without restrictions, thereby increasing the positivity of the experience and improving cognitive ability. These three factors not only enhance the engagement of people with SSD in the training tasks but also deepen and change their level of engagement during training. Such programs allow participants to invest their mental and physical effort without worrying about negative feedback or punishment for errors. We therefore believe that the multitasking nature of video games makes it a suitable training model for the integration of physical and mental abilities, which echoes the suggestion made by a previous study [73] regarding the necessity of involving people with SSD in such programs to promote their physical and mental strength simultaneously.

The main limitation of the study is that it lacked a real control against which to assess the effectiveness of both forms of multitask interventions. Although the cognitive and motor control performance of people with SSD improved after the two stages of training, comprising two different types of multitasks, a real control for the intervention at both the stages was lacking, therefore, the third measure (Po2) might be a results video game training or a continuous improvement following traditional training. Future studies with a control for each type of training tasks should be conducted to confirm the effects of both types of multitasks. Moreover, this study was conducted in one of the three therapeutic communities of a local psychiatric hospital with small sample size. Therefore, the generalizability of the results is limited to the SSD patients with the same pre-training status in this study. Furthermore, appropriate follow-up to monitor the long-term effects is necessary. Moreover, sufficient evidence to support the effects of the training model on fall frequency was lacking. The intervention period in this study was only 6 months, and it was difficult to follow-up fall accident injuries after the study to evaluate the long-term effect of the intervention. Besides, the training effects might be over-estimated because the target group excluded those with fracture injuries; with significant impairments in visual, auditory, and cognitive functions; with high fall risks; and who were seriously bedridden due to fall injuries. What is more is that the long-term effects of the

training program on decrease of fall incidence were not examined. The requiring immediate interventions were not examined. Finally, the reliability of BBT for people with SSD was not examined, which might impede the ability of BBT to demonstrate the change of gross manual dexterity after intervention in this study. Future studies should address the abovementioned limitation of this study to further confirm the effects of TMT and VGMT on cognition and motor control in people with schizophrenia.

## Supporting information

**S1 File.**
(PDF)

**S2 File. CONSORT 2010 flow diagram.**
(PDF)

**S1 Checklist. CONSORT 2010 checklist of information to include when reporting a randomised trial**∗**.**
(PDF)

## Author Contributions

**Conceptualization:** Jen-Suh Chern.

**Data curation:** Jen-Suh Chern, Yu Yang.

**Formal analysis:** Tzu-Yun Chien, Jen-Suh Chern.

**Funding acquisition:** Jen-Suh Chern.

**Investigation:** Tzu-Yun Chien.

**Methodology:** Jen-Suh Chern.

**Project administration:** Jen-Suh Chern.

**Resources:** Jen-Suh Chern, San-Ping Wang, Yu Yang.

**Software:** San-Ping Wang.

**Supervision:** Jen-Suh Chern, San-Ping Wang.

**Validation:** Jen-Suh Chern.

**Writing – original draft:** Tzu-Yun Chien, Jen-Suh Chern, Yu Yang.

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
