## [Decision Letter · Decision Letter 0]

18 Mar 2021

PONE-D-20-28334

Effects of multi-task training on cognition and motor control in patients with schizophrenia spectrum disorders

PLOS ONE

Dear Dr. Chern,

Thank you for submitting your manuscript to PLOS ONE. After careful consideration, we feel that it has merit but does not fully meet PLOS ONE’s publication criteria as it currently stands. Therefore, we invite you to submit a revised version of the manuscript that addresses the points raised during the review process.

We look forward to receiving your revised manuscript.

Kind regards,

Sandra Carvalho

Academic Editor

PLOS ONE

Journal Requirements:

2. Please include your tables as part of your main manuscript and remove the individual files. Please note that supplementary tables should be uploaded as separate "supporting information" files.

Reviewers' comments:

Reviewer's Responses to Questions

**Comments to the Author**

1. Is the manuscript technically sound, and do the data support the conclusions?

Reviewer #1: Yes

Reviewer #2: Partly

Reviewer #3: Partly

2. Has the statistical analysis been performed appropriately and rigorously? 

Reviewer #1: Yes

Reviewer #2: No

Reviewer #3: No

3. Have the authors made all data underlying the findings in their manuscript fully available?

Reviewer #1: No

Reviewer #2: Yes

Reviewer #3: Yes

4. Is the manuscript presented in an intelligible fashion and written in standard English?

Reviewer #1: Yes

Reviewer #2: Yes

Reviewer #3: No

5. Review Comments to the Author

Reviewer #1: Thank you for the opportunity to read the article, “Effects of multi-task training on cognition and motor control in patients with schizophrenia spectrum disorders.” The issue is certainly important in terms of promoting the population of people with SCCD. The research is necessary and deals with very relevant areas in treating populations with cognitive and motor difficulties. It should be noted that the very fact that the participants completed the entire study – which is long and consists of two stages – and persevered to arrive is an achievement. Well done.

I have significant comments to make the article clearer. I upload reviewers attachment with specific comments to the authors

Regards the data underlying the findings - I didn't see them but maybe they are exist.

.

Reviewer #2: The manuscript entitled ‘Effects of multi-task training on cognition and motor control in patients with schizophrenia spectrum disorders’ with the aim to examine the effects of traditional multi-task (MT) training and exergames as a new form of MT on cognition and motor control performance in patients with Schizophrenia spectrum disorder (SCCD).

The manuscript requires further improvement based on the comments below.

Methods

Line 182-183, the sentence ‘The data of those who dropped out after the stage 1 training was expelled from the analysis.’ not clear. In Figure 1, it was stated n=25 were analyzed.

Line 183-184, more information for the sample size calculation to be provided i.e. outcome measures, mean/sd or %, 1 or 2-tailed test etc.

Line 276 & 280, italicized r or r to be standardized.

Statistical analysis

Line 360, exact name for the type of repeated measures ANOVA to be stated.

Line 361, proper citation of SPSS including publisher name to be stated.

Line 362, 0.058 ≦ ηp2 and 0.058 ≦ ηp2 < 0.138 requires revision and include 0.058 ≥ ηp2.

Type of post hoc comparison and adjustment to multiple comparison to be stated.

Results

Table 1, mean ± sd to be displayed first before ranges. Ranges to be denoted in the footnote as minimum to maximum value. Typo error for age figures 59.6111.46. Symbol ~ and unit Kgw to be replaced with dash (-) and kg respectively.

Table 2, N for Pre, Po1 and Po2 to be stated. What the symbol a, b, c refers to be denoted in the table footnote and in superscript form in table. Post hoc comparison of group and time to be clearly denoted. Technically, p value cannot be zero (.000) (to use symbol < and denoted with symbol ***). Decimal points for the figures to be standardized. Likewise, with Table 3.

Line 372, for CTT1T, CTT2T, T to be separated/spaced out.

Line 382, 388, 395, the figures to be cited from table.

The results could be further explored by stratifying according to age group and gender.

Some references did not comply to the journal format.

Clinical trial registration number to be stated in the manuscript once received.

Reviewer #3: Dear Authors,

The manuscript focuses on important, under investigated issue of motor functioning among people with schizophrenia. However, there are two major concerns with this study. First, there is no theoretical background for definition of the first provided intervention as multitasking. Within general classification of the interventions, the first intervention is a classic motor training, without multitasking. The intervention with the computer games somehow can fall under the definition of multitasking, but also need to be explained. Second, the study design provide a base to do some conclusion about effectiveness of motor training in general, with little ability to compare between two types of the interventions. The findings somehow support effectiveness of the traditional motor training, but did not support any conclusion about the effectiveness of videogames training.

There are some additional issues, such as properness of the statistical analysis and English spelling. I strongly recommend addressing these issues first, followed by reorganization of the discussion sections in order to consider this manuscript for the publication.

Some detailed comments are attached bellow.

Abstract:

Line 47: Please add standard deviation of the age.

Line 48: TMT – you use this acromion first time. Please spell it.

Introduction

The definition of multitasking is missing.

Line 66: “neuropsychological and adaptive motor behavior function” The difference between the neuropsychological resources and motor adaptive resources is not clear. In order to perform motor adaptation we need neuropsychological functions. The resources may be presented using Canadian model: cognitive/neurological, emotional, motor, etc.

Line 67: Replace the word “responsibilities” with “role”. Additional editing for proper English is needed through the manuscript.

Line 71: “satisfying people's pursuit of the quality of life” should be replace with “enabling to obtain quality of life” ;

“Sharpening” should be replace with “amelioration”

Lines 73-78: The phrasing makes it difficult to understand your explanation of the multitasking. Each point should be presented separately and further explained. In addition, the references should be added.

Lines 80-84: Please add example of multitasking intervention, describing it in brief.

Lines 85-86: The reference should be added

Lines 87-88: Hallucinations and delusions are not conceptualized currently as cognitive disorders. Please rephrase.

Lines 92: Please avoid usage with “patients with SSD “ and use “people with SSD” instead.

Line 94: “ The collective deficits” - not clear what you mean, cumulative?

Lines 95-96: Please add prevalence of falls in people with schizophrenia and age-specific characteristics in comparison to healthy controls.

Line 114: MMSE – should be spelled out

Line 115: MMSE less sensitive to difficulties with executive functions. Please address this issue.

Lines 134-135: The last part of the sentence is not clear. Please rephrase to support explicitly the rational of your study.

Lines 143-144: “Moreover…” Provide details on the improvement in the emotional state and add the reference.

Line 147: The name of software for training should be added.

Line 184: It is not clear how the sample size was calculated. What basic data was used to do the calculation? What sample size you find? If the calculation was done a priory or after the study completion? How it is related to your actual sample size in the study.

Line 190: “posttest: do you mean post-training?

Lines 200-226: The intervention that you provide at the first stage can be hardly considered as multitasking. You describe the standard motor training, but not multitasking. Please provide strong theoretical base for your claim that this intervention is a multitasking. The intervention did not involve simultaneously several tasks or several sensory systems.

Results – Please provide information on data distribution. If the distribution is different from normal, you should present the data with median and inter-quartile range, since mean and SD are not appropriate. Moreover, if the data was not distributed normally, bootstrapping should be performed before applying repeated measures ANOVA.

Discussion

The main limitation of the study design that you had no real control for effectiveness of any intervention. In other words, the third measure (Po2) may be either a result of video-game training or a continuous improvement following traditional motor training. This point should be clearly presented in the manuscript.

You use in discussion the term video-game first time. This is good definition for the second intervention. Please replace with this term the term “exergames” that you used previously.

There is a lot of information through the discussion that should be moved to introduction section (e.g. lines 411-418). Please focus the discussion on your findings.

CTT is not multitasking test, CTT-B require cognitive shifting. Please be more specific with your writing. “CTT … as an importance outcome measure of multitasking abilities.” In addition you should rephrase your statements for proper English.

Line 442: “clinical effort devoted to upper limb motor function training in SSD has not been reported, yet” This point is not clear.

Lines 490-492: Please rephrase your conclusion since the study design did not suppotd this statement.

6. PLOS authors have the option to publish the peer review history of their article (what does this mean?). If published, this will include your full peer review and any attached files.

Reviewer #1: No

Reviewer #2: No

Reviewer #3: No

---

## [Author Response · Author response to Decision Letter 0]

29 Jun 2021

Please refer to the "Response to Reviews" file.

---

## [Decision Letter · Decision Letter 1]

20 Aug 2021

PONE-D-20-28334R1

Effects of multitask training on cognition and motor control in patients with schizophrenia spectrum disorders

PLOS ONE

Dear Dr. Chern,

Thank you for submitting your manuscript to PLOS ONE. After careful consideration, we feel that it has merit but does not fully meet PLOS ONE’s publication criteria as it currently stands. Therefore, we invite you to submit a revised version of the manuscript that addresses the points raised during the review process.

We look forward to receiving your revised manuscript.

Kind regards,

Sandra Carvalho

Academic Editor

PLOS ONE

Journal Requirements:

Reviewers' comments:

Reviewer's Responses to Questions

**Comments to the Author**

1. If the authors have adequately addressed your comments raised in a previous round of review and you feel that this manuscript is now acceptable for publication, you may indicate that here to bypass the “Comments to the Author” section, enter your conflict of interest statement in the “Confidential to Editor” section, and submit your "Accept" recommendation.

Reviewer #1: All comments have been addressed

Reviewer #2: (No Response)

Reviewer #3: All comments have been addressed

2. Is the manuscript technically sound, and do the data support the conclusions?

Reviewer #1: Yes

Reviewer #2: Partly

Reviewer #3: Partly

3. Has the statistical analysis been performed appropriately and rigorously? 

Reviewer #1: Yes

Reviewer #2: (No Response)

Reviewer #3: Yes

4. Have the authors made all data underlying the findings in their manuscript fully available?

Reviewer #1: Yes

Reviewer #2: Yes

Reviewer #3: Yes

5. Is the manuscript presented in an intelligible fashion and written in standard English?

Reviewer #1: Yes

Reviewer #2: Yes

Reviewer #3: Yes

6. Review Comments to the Author

Reviewer #1: The authors corrected the article in accordance with the comments. Thank you very much for the opportunity to evaluate the article

Reviewer #2: Table 3, a, b, c to be denoted in the table footnote. n for Pre, Po1, Po2 to be stated.

Line 430, adjustment with multiple comparisons, Bonferroni to be stated.

Figure 2. what is the small box beside re-training measurement?

Reviewer #3: Dear Authors,

Thank you for addressing my previous comments. The manuscript presents in much more accurate way the study. I have some additional comments to the revised version of the manuscript.

Abstract:

Line 54: Please re-phrase the results report on effectiveness of “both” interventions according to the study design. You actually can conclude about the first intervention effectiveness and about combination of the first and second together.

Introduction

Line 69: “The neuropsychological system..” Please elaborate on connection between this sentence and the following one. Reference for this statement should be added.

Lines 74-75: “Additionally….” Please check wording of the sentence. It sounds less logical.

Line 85: “balance” – please replace with balance

Line 109: double “and”. Please delete one of them.

Lines 119 - 120 : “Impairments in cognitive functions usually occur during the first SSD episode” Please add more updated references.

Lines 124-127: “scores on the Minimal 125 Mental Status Evaluation (MMSE) … functions” should be removed to the discussion section.

Lines 137-141: “. Furthermore, Hidese and… dysfunction” is less relevant for the study. Please delete from the manuscript. You can provide direct references on brain functioning alteration in schizophrenia. It is well established factor.

Lines 177-179: “Therefore, the purpose of this study was to examine the differential effects 178 between traditional goal-directed multitasking activities and video games on cognition and 179 upper extremity and postural control in people with SSD.” The goal should be rephrase since the study design not allow you to draw conclusion on differentiation between two types of the training.

Lines 179-182: You should rephrase the sentence according with the understanding of your study design.

Measurements

Lines 318-319: Please remove the report on the blinding group allocation since there is no groups in the study.

Results

Line 466: “indicated that VGMT further improved TUG scores significantly…” Please state clear that it is “VGMT training following TMT training”. There is no group with VGMT training only or first, so you have no data to relay on concluding about the VGMT training only.

Line 469: “especially the form of video games” The same comment as previous. Please rephrase.

Line 476: “multitasking in the form of video game” – The same as previous. You can speak about combination of two interventions. Please rephrase you reports throughout the manuscript.

Discussion

Lines 498 – 504: The same sentence duplicated twice. Please remove one of them.

Line 509: “Therefore, ….” – please reword the sentence since it is not clear . “aht” - should be deleted.

Lines 513-516: “The difficulty faced by people with SSD in multitasking was independent of the deficits in general processing speed, working memory, and attention.” You should support this statement with statistic data. Please add statistical support for your statement.

Lines 515-517: “Conversely, multitasking difficulties …” the same comments as the previous one.

Lines 536-537: “Hidese and colleagues [20] … impairment and decreased insular volume and white matter in people with SSD [17]. “ Please delete this sentence since it has little contribution to the topic of the manuscript.

Please add to the limitations that you have small sample size.

7. PLOS authors have the option to publish the peer review history of their article (what does this mean?). If published, this will include your full peer review and any attached files.

Reviewer #1: No

Reviewer #2: No

Reviewer #3: No

---

## [Author Response · Author response to Decision Letter 1]

11 Nov 2021

We thank the Editor's comments. The funding statement in the Title Page has been removed and there is no any funding-related text in the rest of the manuscript except in the title page. The funding statement that we would like to be shown in the online submission is stated in the cover letter as the Editor's suggestion. 

One typo “manisfest” in line 70 is corrected as “ manifest” and one duplicated word “the” in line 233 is removed as shown in the Revised Manuscript w/tracked changes. The Revised Manuscript w/tracked changes in PDF file format and the revised Manuscript in .docx format are uploaded. 

All the authors thank the Editor's comments and wish the correction fulfilling the Editor's comments.

---

## [Editor Report · Decision Letter 2]

9 Dec 2021

PONE-D-20-28334R2

Effects of multitask training on cognition and motor control in people with schizophrenia spectrum disorders

PLOS ONE

Dear Dr. Chern,

Thank you for submitting your manuscript to PLOS ONE. After careful consideration, we feel that it has merit but does not fully meet PLOS ONE’s publication criteria as it currently stands. Therefore, we invite you to submit a revised version of the manuscript that addresses the points raised during the review process.

Please find below some comments from the reviewers, which I believe will improve the current manuscript (please see additional Editor comments).

We look forward to receiving your revised manuscript.

Kind regards,

Sandra Carvalho, Ph.D.

Academic Editor

PLOS ONE

Journal Requirements:

Additional Editor Comments (if provided):

Reviewer # 2

Table 3, a, b, c to be denoted in the table footnote. n for Pre, Po1, Po2 to be stated. Line 430, adjustment with multiple comparisons, Bonferroni to be stated. Figure 2. what is the small box beside re-training measurement?

Reviewer # 3

Dear Authors, Thank you for addressing my previous comments. The manuscript presents in much more accurate way the study. I have some additional comments to the revised version of the manuscript. Abstract: Line 54: Please re-phrase the results report on effectiveness of “both” interventions according to the study design. You actually can conclude about the first intervention effectiveness and about combination of the first and second together. Introduction Line 69: “The neuropsychological system..” Please elaborate on connection between this sentence and the following one. Reference for this statement should be added. Lines 74-75: “Additionally….” Please check wording of the sentence. It sounds less logical. Line 85: “balance” – please replace with balance Line 109: double “and”. Please delete one of them. Lines 119 - 120 : “Impairments in cognitive functions usually occur during the first SSD episode” Please add more updated references. Lines 124-127: “scores on the Minimal 125 Mental Status Evaluation (MMSE) … functions” should be removed to the discussion section. Lines 137-141: “. Furthermore, Hidese and… dysfunction” is less relevant for the study. Please delete from the manuscript. You can provide direct references on brain functioning alteration in schizophrenia. It is well established factor. Lines 177-179: “Therefore, the purpose of this study was to examine the differential effects 178 between traditional goal-directed multitasking activities and video games on cognition and 179 upper extremity and postural control in people with SSD.” The goal should be rephrase since the study design not allow you to draw conclusion on differentiation between two types of the training. Lines 179-182: You should rephrase the sentence according with the understanding of your study design. Measurements Lines 318-319: Please remove the report on the blinding group allocation since there is no groups in the study. Results Line 466: “indicated that VGMT further improved TUG scores significantly…” Please state clear that it is “VGMT training following TMT training”. There is no group with VGMT training only or first, so you have no data to relay on concluding about the VGMT training only. Line 469: “especially the form of video games” The same comment as previous. Please rephrase. Line 476: “multitasking in the form of video game” – The same as previous. You can speak about combination of two interventions. Please rephrase you reports throughout the manuscript. Discussion Lines 498 – 504: The same sentence duplicated twice. Please remove one of them. Line 509: “Therefore, ….” – please reword the sentence since it is not clear . “aht” - should be deleted. Lines 513-516: “The difficulty faced by people with SSD in multitasking was independent of the deficits in general processing speed, working memory, and attention.” You should support this statement with statistic data. Please add statistical support for your statement. Lines 515-517: “Conversely, multitasking difficulties …” the same comments as the previous one. Lines 536-537: “Hidese and colleagues [20] … impairment and decreased insular volume and white matter in people with SSD [17]. “ Please delete this sentence since it has little contribution to the topic of the manuscript. Please add to the limitations that you have small sample size.
---

## [Author Response · Author response to Decision Letter 2]

30 Dec 2021

Please refer to the file with the name of "Response to Reviewers" for responses to each comments of reviewer#2 and #3.

---

## [Editor Report · Decision Letter 3]

17 Feb 2022

Effects of multitask training on cognition and motor control in people with schizophrenia spectrum disorders

PONE-D-20-28334R3

Dear Dr. Chern,

We’re pleased to inform you that your manuscript has been judged scientifically suitable for publication and will be formally accepted for publication once it meets all outstanding technical requirements.

Kind regards,

Sandra Carvalho, Ph.D.

Academic Editor

PLOS ONE

---

## [Editor Report · Acceptance letter]

21 Feb 2022

PONE-D-20-28334R3 

Effects of multitask training on cognition and motor control in people with schizophrenia spectrum disorders 

Dear Dr. Chern:

I'm pleased to inform you that your manuscript has been deemed suitable for publication in PLOS ONE. Congratulations! Your manuscript is now with our production department. 

Kind regards, 

on behalf of

Professor Sandra Carvalho 

Academic Editor

PLOS ONE